# Assessment of particle size magnifier inversion methods to obtain particle size distribution from atmospheric measurements

Tommy Chan[1,2], Runlong Cai[1,2], Lauri R. Ahonen[2], Yiliang Liu[3], Ying Zhou[1], Joonas Vanhanen[4], Lubna Dada[1,2], Yan Chao[1,2],

Yongchun Liu[1], Lin Wang[3], Markku Kulmala[1,2], Juha Kangasluoma[1,2]

[1]Aerosol and Haze Laboratory, Beijing Advanced Innovation Center for Soft Matter Science and Engineering, Beijing University of Chemical Technology, Beijing 100029, China
[2]Institute for Atmospheric and Earth System Research / Physics, Faculty of Science, University of Helsinki, Helsinki 00014 Finland
[3]Shanghai Key Laboratory of Atmospheric Particle Pollution and Prevention (LAP[3]), Department of Environmental Science and Engineering, Fudan University, Shanghai 200433, China.
[4]Airmodus Ltd., Erik Palménin aukio 1, Helsinki 00560, Finland

*Correspondence to*: Tommy Chan (tommy.chan@helsinki.fi), Juha Kangasluoma (juha.kangasluoma@helsinki.fi)

**Abstract.** Accurate measurements of the size distribution of atmospheric aerosol nano-particles are essential to the understanding of new particle formation and growth. This is particularly crucial at the sub-3 nm range because of the growth of newly-formed nano-particles. The challenge in recovering the size distribution is due its complexity and the fact that not many instruments currently measure at this size range. In this study, we used the particle size magnifier (PSM) to measure atmospheric aerosols. Each day was classified into one of the three event types: new particle formation (NPF) event, non-event and haze event. We then compared four inversion methods (step-wise, kernel, Hagen and Alofs and expectation-maximization) to determine their feasibility to recover the particle size distribution. In addition, we proposed a method to pre-treat measured data and introduced a simple test to estimate the efficacy of the inversion itself. Results showed that all four methods inverted NPF events well; but the step-wise and kernel methods fared poorly when inverting non-event and haze events. This was due to their algorithm, such that when encountering noisy data (e.g. air mass fluctuations or low sub-3 nm particle concentrations) and under the influence of larger particles, these methods overestimated the size distribution and reported artificial particles during inversion. Therefore, using a statistical hypothesis test to discard noisy scans prior to inversion is an important first step to achieve a good size distribution. As a first step after inversion, it is ideal to compare the integrated concentration to the raw estimate (i.e., the concentration difference at the lowest supersaturation and the highest supersaturation) to ascertain whether the inversion itself is sound. Finally, based on the analysis of the inversion methods, we provide procedures and codes related to the PSM data inversion.

## 1 Introduction

Gas-to-particle conversion proceeds via molecular clustering and subsequent cluster growth in various systems, such as atmospheric particle formation events, combustion processes or nano-particle synthesis (Almeida et al., 2013; Carbone et al., 2016; Fang et al., 2018; Feng et al., 2015; Jokinen et al., 2018; Kulmala et al., 2004; Sipilä et al., 2016). Particle growth occurs

at the size of a few nanometers, and direct measurements are presently available to probe the dynamics of the process. Instruments such as the scanning mobility particle sizer (DEG-SMPS; Jiang et al. (2011a)), particle size magnifier (PSM, Airmodus Ltd., Finland; Vanhanen et al. (2011)), neutral cluster and air ion spectrometer (NAIS, Airel Ltd., Estonia; Mirme and Mirme (2013)) or the pulse height analysis condensation particle counter (PHA CPC; Marti et al. (1996)) have been previously applied to directly measure the formation and growth of the clusters (Cai et al., 2017; Jiang et al., 2011b; Kontkanen

et al., 2017; Manninen et al., 2010; Sipilä et al., 2009; Yu et al., 2017). These instruments have different operation principles and instrument functions, and therefore need specific data inversion methods to obtain a reliable conversion from the measured (i.e., raw) data to a particle number size distribution. The SMPS, for instance, is a differential method, measuring a narrow size band at one time (Stolzenburg and McMurry, 2008). The PSM, on the other hand, is a cumulative method, which measures total particle concentrations above certain threshold diameters (Cai et al., 2018). The comparison of the size distributions

measured by these and some other instruments reveal that there is still work required to improve the accuracy of the measured sub-10 or sub-3 nm size distributions . Our focus in this study is the data inversion of the PSM for applications in atmospheric measurements.

Particle detection in the PSM is based on condensational growth of particles in two separate stages. In the first stage, the

particles are grown with diethylene glycol (DEG) up to around 100 nm by mixing heated DEG vapour with a sample flow. In the second stage, the activated particles are further grown with butanol. The cut-off diameter (i.e., the diameter at which 50% of the particles are activated in the first stage) varies approximately between 1-3 nm, depending on the mixing ratio of DEG vapour. The mixing ratio is controlled by varying the flow rate that is saturated by DEG. Therefore, the raw data for the inversion problem is the measured total particle concentration above a certain cut-off diameter as function of the flow rate

through the saturator. Several parameters need to be considered in this specific inversion problem: 1) the shape of the cut-off curves (instrument size function); 2) the data pre- and post-treatment to minimize random noise in the data; and 3) the mathematical method for the inversion.

To retrieve the sub-3-nm aerosol size distributions from the PSM raw data, the step-wise method and the kernel function

method (Lehtipalo et al., 2014) have been used for PSM data inversion of atmospheric measurements. The step-wise method is the promoted inversion method for commercial PSMs and supported by Airmodus. It neglects the impact of the limited size resolution of PSM to the measured aerosol concentration at each saturation flow rate, and hence, causes systematic biases. The kernel function method considers the finite size resolution during inversion; however, it is more sensitive to random uncertainties than the step-wise method. To improve PSM inversion, Cai et al. (2018) compared four inversion methods: the

step-wise method, the kernel function method, the Hagen-Alofs (H&A) method (Hagen and Alofs, 1983), and the expectation-

maximization (EM) algorithm (Hagen and Alofs, 1983; Maher and Laird, 1985). It was suggested that the EM algorithm considers the finite size resolution and is less sensitive to random errors compared with the kernel function and H&A methods.

However, the study by Cai et al. (2018) was mainly based on theoretical simulations and well-controlled laboratory experiments. For real atmospheric measurements, the larger measurement uncertainties compared to laboratory experiments may pose a challenge for each of these inversion methods. As indicated in Cai et al. (2018), the random uncertainty of the inverted size distribution is significant even under a relative uncertainty of 10% in the raw data. Furthermore, different from lab experiments in which the detection efficiency for each particle size is known, the PSM detection efficiency of atmospheric aerosols are not determined due to their unknown chemical compositions. This unknown detection efficiency may also cause non-negligible biases (Kangasluoma and Kontkanen, 2017) to the inverted PSM data. As a result, the feasibility and performance of these inversion methods need further verification and testing using the atmospheric measured data.

In this study, we present the four methods to invert measured, atmospheric PSM data obtained in Beijing, China. We discuss the following aspects to obtain the particle size distribution: 1) the usability of individual scans; 2) the comparison of typical, inverted individual scans using the four inversion methods; 3) the characteristics of each inversion method when applied to atmospheric data; and 4) a simple method to determine, as a first approximate, the reliability of the inversion. Finally based on the analysis of the performance of the inversion methods, we provide recommendations on how to invert atmospheric data measured with the PSM.

## 2 Materials and Methods

### 2.1 Site Description

The study site is located on the fifth floor at the Aerosol and Haze Laboratory of the Beijing University of Chemical Technology, located at the Haidan District in Beijing, China (39° 56' 31" N, 116° 17' 49" E, and 58 m above sea level). The laboratory is near the 3[rd] ring road of Beijing and gives a good representation of an urban environment that is surrounded by traffic roads, highways, residential and commercial buildings. The combination of these different zones brings together pollution from local (e.g. traffic emissions, cooking) and neighbouring sources.

This study was conducted between 15 Jan. and 31 Mar. 2018 ($n = 76$ days) and was representative of a Beijing winter. Beijing winters are generally cold and dry, with an average temperature of 0°C. The average monthly temperature highs are 2°C, 5°C and 12°C, and monthly lows are -9°C, -6°C and 0°C in January, February and March, respectively. During these three months, the overall average humidity and rainfall was ~44% and 5.33 mm, respectively.

## 2.2 Classification of event types

Three event types were identified for the study: new particle formation (NPF), haze and non-event (i.e. neither haze nor NPF events). An NPF event is classified according to the method introduced by Dal Maso et al. (2005), such that the growth of particles increase in size across different modes over several hours. Haze events were identified as days when the relative humidity was lower than 80% and the visibility range was less than 10 km – for a duration of 12 continuous hours. During the study period, we observed in total, 29 NPF, 36 haze and 11 non-events. NPF events were typically isolated as daily events that occurred after sunrise and continued to the early afternoon. Meanwhile, haze events occurred randomly throughout the day and could last for several days. These three event types did not commonly overlap one another during the study period.

## 2.3 Aerosol particle measurements

Aerosol particle number concentration, expressed in $\#/cm^3$, was measured using a butanol-based condensation particle counter (CPC; model A20, Airmodus Ltd., Finland). The CPC can measure a maximum particle concentration of up to $10^5 \#/cm^3$. The CPC is connected directly to the Particle Size Magnifier (PSM; model A10, Airmodus Ltd., Finland). The PSM is a pre-conditioner for the CPC that uses diethylene glycol as the working fluid to activate and grow nano-sized particles (~1-3 nm) so that they can be detected with the CPC (Vanhanen et al., 2011). A 1.3 m long horizontal inlet from where the aerosol particles entered was fixed to the PSM inlet and a core sampler was fitted to reduce sampling line losses (Fu et al., 2019; Kangasluoma et al., 2016). Losses due to particle diffusion, penetration and core sampling were accounted for after the data inversion. If the sampling is done well (e.g., using a core sampler at the PSM inlet), the line losses can be negligible (Fu et al., 2019; Kangasluoma et al., 2016). If the losses are non-negligible but not large enough to decrease line penetration close to 0, the line losses can be corrected after the inversion for the size-classified data e.g., using the size bin mean diameter. To maintain brevity, the term PSM will be used henceforth to refer to the PSM or the combination of CPC and PSM.

The PSM measures the total particle concentration by mixing sample aerosol flow with heated saturated flow containing diethylene glycol. By varying the saturator flow rate, the mixing ratio of the sample flow and saturated flow changes, thus particle cut-off size can be changed. In other words, particles of specific diameters assuming constant composition will be activated and grow to larger sizes based on the mixing ratio. In practice, the PSM can operate by scanning (i.e., incrementing followed by decrementing continuously) the saturator flow from 0.1-1.3 liters per minute (lpm) to vary the particle cut-off size. For a constant particle size, the detection efficiency of the PSM as a function of the saturator flow rate is close to a sigmoid function, for which inversion methods taking into account the instrument function is needed. In this study, we adjusted the duration of each scan to 240 seconds, recording data at one-second intervals.

## 2.4 Data pre- and post-treatment

During the process of converting the measured data into a particle size distribution, the data were checked and treated prior to inversion (pre-treatment) and following inversion (post-treatment). The programming language used for all data handling and data analyses was MATLAB ver. R2019a (The Mathworks, Inc.). Because of fluctuations in the air masses, the measured concentration as a function of the supersaturation is not always monotonically increasing, making the inversion procedure mathematically unsound. During periods when sub-3 nm particles are low, theoretically the measured concentration should be relatively constant as a function of the supersaturation. But near the detection limit of the PSM, the inversion may face problems, which we will be explain in this study. Meanwhile, when the particle concentration is high, it is very possible that the concentration is real (Kangasluoma et al., 2020). Therefore, it is sensible to discard any scans not showing a positive correlation between the supersaturation and measured concentration to avoid inversion of any artificial counts from scans when there is clearly no sub-3 nm particles present – or if their presence is dubious.

The pre-treatment included a data quality check and noise removal procedure. As there is a general, near-linear relationship between the saturator flow rate and measured concentration, the quality check employed a statistical hypothesis test (Spearman's rank correlation coefficient) for each scan that retained scans considered significant and positive, while discarding scans considered contrary to retained scans. A statistical significance was set at $p < 0.05$ to consider subtle changes, which could be a real atmospheric influence. Following the significance test, a locally weighted scatterplot smoothing filter (LOWESS) was used over a span of six seconds for each single scan. The purpose of the smoothing was to minimize fluctuations or noise – for example, due to sudden changes in air mass. We explored the performance of the pre-treatment quality scan, especially pertaining to what scans were retained and discarded. In addition, we applied each inversion method to these retained and discarded scans to further understand the inversion process. A smoothing average over two scans (i.e. eight minutes in this study) was applied after the inversion. This reduced the random uncertainty in the inverted data and facilitates, for example, the calculation of particle growth and formations rates. Note that choosing another noise filtering mechanism (e.g., median filtering) can be used, depending on the user's discretion. The smoothing is done after rather than before the inversion because the measured concentration is autocorrelated, whereas the inverted size distribution is simply a function of particle diameter and, therefore, can be averaged for the same size bin.

## 2.5 Data inversion

In this study, four inversion methods were tested using the data obtained in atmospheric measurements: the step-wise method (Lehtipalo et al., 2014), kernel function method (Lehtipalo et al., 2014), Hagen and Alofs method (H&A; Hagen and Alofs (1983)) and the expectation and maximization algorithm (EM; Dempster et al. (1977); Maher and Laird (1985)).

165

The particle number concentration measured with the PSM uses the Fredholm integral equation of the first kind to determine the particle size distribution:

$$R_i = \int_0^{+\infty} \eta\left(s_i, d_p\right) \times n \times \mathrm{d}d_p + \varepsilon_i, \tag{1}$$

170

where $R_i$ is the raw concentration for a saturator flow rate of $s_i$; $\eta$ is the detection efficiency calculated from $s$ and $d_p$; $d_p$ is the particle size; $n(\mathrm{d}d_p)$ is the particle size distribution function (units: particles/cm$^3$/nm$^1$); and $\varepsilon_i$ are the errors in the measurement at $s_i$. For atmospheric measurements, the relatively large $\varepsilon_i$ poses a challenge to data inversion. For example, the detection efficiency being large, there is measurement background from the PSM and measured concentrations that are small can contribute to these errors.

175

The step-wise method is currently the proprietary inversion method for use with the PSM. When calculating particle size distributions using the step-wise method, the size resolution of the PSM is assumed to be infinite (i.e., the kernel function is approximated with a Dirac delta function whose area is equal to the real kernel but height is infinite). Based on this assumption, it can be demonstrated that there is a one-to-one relationship between the saturator flow rate and the activated particle diameter, hence the particle number concentration in the specific size range can be obtained by calculating the measured particle number concentration increment (after correcting the detection efficiency) in its corresponding saturator flow rate range. The expression for the stepwise method, introduced in Lehtipalo et al. (2014), in practical use is:

$$n_\mathrm{m} = \frac{2(R_{i+1} - R_i)}{\eta(s_\mathrm{max}, d_i) + \eta(s_\mathrm{max}, d_{i+1})} \times \frac{1}{d_i - d_{i+1}}, \tag{2}$$

185

where $n_\mathrm{m}$ is the particle size distribution (d$N$/d$d_\mathrm{m}$) at diameter $d_\mathrm{m}$; $d_\mathrm{m}$ is the median diameter of $d_i$ and $d_{i+1}$; $d_i$ and $d_{i+1}$ are the corresponding diameter of saturator flow rates $s_i$ and $s_{i+1}$, respectively, and this one-to-one relationship is obtained based on the infinite-resolution assumption; $R_i$ and $R_{i+1}$ are the raw concentration recorded by the PSM and dilution has been corrected; $s_\mathrm{max}$ is the maximum saturator flow rate; and $\eta$ is PSM detection efficiency at the given saturator flow rate and particle diameter. The inverted d$N$/d$d_\mathrm{m}$ was later converted into d$N$/dlog$d_\mathrm{m}$. The derivation of the step-wise method can be seen in the Supplementary Information.

The kernel function and H&A methods both account for the kernel functions of the PSM. At each saturator flow rate, the measured total particle number concentration (or its derivative with respect to the saturator flow rate) is equal to the sum of particle number concentrations in each size bin multiplied by their detection efficiencies (or correspondingly kernel functions). The particle number concentrations in each size bin are obtained by solving the non-homogeneous, linear equations that relate saturator flow rates and particle number concentrations. The difference between the kernel function and the H&A methods is the number of assumed particle size bins. The kernel function method uses a size bin number (typically four to six) much less than the number of saturator flow rates, while the H&A method uses a size bin number (theoretically infinite) much more than the number saturator flow rates and then reduces the size bin number to the saturator flow rate number using predetermined interpolation functions. Note that the H&A method itself does not specify that either the detection efficiencies or the kernel functions should be used for data inversion. In this study, detection efficiencies are used in the H&A method to avoid any uncertainties introduced when estimating the derivate of particle number concentration with respect to saturator flow rate; and to keep in accordance with Cai et al. (2018).

The EM algorithm is an iterative algorithm based on the theories of probability, which is used in the inversion of diffusion batteries (Maher and Laird, 1985; Wu et al., 1989) and machine learning (e.g., Erman et al., 2006). The expressions for the EM algorithm are:

$$R_{i,j} = \frac{n_j \times \eta(s_i, d_j) \times \Delta d_j}{\sum_{j=1}^{J} n_j \times \eta(s_i, d_j) \times \Delta d_j},$$
(3)

$$n_j = \frac{\sum_{i=1}^{I} R_{i,j}}{\sum_{I=1}^{I} \eta(s_i, d_j) \times \Delta d_j},$$
(4)

where $I$ is the total number of saturator bins and the $i^{th}$ saturator flow rate is $s_i$; $J$ is the total number of particle size bins and the $j^{th}$ particle size is $d_j$; $\Delta d_j$ is the width of the $j^{th}$ size bin; $n_j$ is the particle size distribution at $d_j$ ($dN/dd_j$); $\eta$ is the PSM detection efficiency for the given $s_i$ and $d_j$; $R_{i,j}$ is the contribution of the $j^{th}$ size bin to the total raw concentration ($R_i$) measured at $s_i$ and it is a latent variable that cannot be directly measured. Similar to the H&A method, $J$ should theoretically be infinite to avoid integral error caused by limited number of size bins and it is practically determined as 50 in this study. For additional details on the four inversion methods, refer to previous studies (Cai et al., 2018; Hagen and Alofs, 1983; Lehtipalo et al., 2014; Maher and Laird, 1985).

**2.6 Data Analysis**

With the study duration amounting to 76 days, we selected a total of 12 days for in-depth analysis: four days with an NPF event, four haze days and four non-event days. For the convenience of comparison, the aerosol size distributions from all the inversion methods are reported in six and eleven size channels, indicated by 7 and 12 limiting diameters of the size channels, respectively. The 6-channel distribution consisted of the following size range (nm): 1.2, 1.3, 1.5, 1.7, 2.0, 2.5 and 2.8 and the

shape of the kernel was approximated with the Gaussian distribution, based on the calibration file (see section 3.1). The 11-channel distribution consisted of the following size range (nm): 1.2, 1.3, 1.4, 1.5, 1.6, 1.7, 1.8, 1.9, 2.0, 2.1, 2.5 and 2.8. The 11-channel inversion was shown to be very similar to the 6-channel inversion (see SI, Fig. S1); and for the exception of the illustration of data inversions of single scans (i.e., Fig. 1), the 6-channel distribution was used in this study as it is a commonly-used size bin range and that it is with certainty that the six kernel function peaks do not significantly overlap with each other

(see SI, Fig. S2). For the step-wise method, the total particle concentrations measured at seven saturator flow rates were inverted into aerosol size distributions at six particle sizes using Eq. (2). For the kernel function method, the measured particle concentration as a function of saturator flow rate was inverted to six size bins using the least square method. For the H&A method and the EM algorithm, the size distribution was firstly inverted into 50 size channels and then reduced to 6 channels by merging adjacent channels. Assuming that there is no error or uncertainty in the kernel functions and particle aerosol number

concentration recorded by the PSM, the inversion methods should be able to distinguish more particle size channels even if their kernel function peaks may overlap with one another or if size resolution is limited. However, considering the atmospheric instability and its particle composition is unknown, we report the size distributions in six channels in this study.

The general challenge in the current sub-3 nm atmospheric size distribution measurements is that there is no real reliable

reference to compare size distributions. In some recent experiments, there has been a concurrent SMPS-based measurement with the PSM (e.g., Kangasluoma et al. (2020)); however, it only gives another independent estimate for the size distribution. Therefore, as a basis of comparison between each inversion method, we compared the integrated total concentration from the inverted distribution to the estimated raw concentration between the mobility diameters of 1.2 and 2.8 nm, $R_{1.2\text{-}2.8}$. This is calculated as the difference between the total particle concentrations measured at the lowest and highest saturator flow rate

(i.e., 0.1 lpm and 1.3 lpm). From this comparison, $R_{1.2\text{-}2.8}$ should be approximately equal to the concentration integrated in the same size range from the inverted size distribution. While the comparison is not expected to yield exact quantitative agreement, it gives an idea of whether the inverted concentration is reasonable, especially during periods when little or no sub-3 nm particles are present. Moreover, this comparison can ensure that the data is internally consistent.

As an additional analysis, we performed a signal-to-noise ratio calculation to determine the detection limit of the PSM from the four inversion methods. In short, the calculation allows the user to identify where the possible noise and measured concentration converge. This is especially important to identify the smallest concentration the inversions allows (which varies between PSM instruments and sample site). The signal-to-noise ratio is calculated by dividing the integrated concentration

from the whole size range with the total concentration measured at the cut-off at 0.1 lpm. It is important to note, however, that

the term "noise" is difficult to define as such, since noise could arise from the data or instrument itself.

The general workflow to obtain the particle size distribution using the PSM was as follows:

1) Determine whether the estimated kernel function curves are reasonable
2) Pre-treat the data to remove scans with no statistical significance
3) Select a filtering method to remove random noise in the measurements
4) Invert the measurement data
5) Correct data for losses
6) Apply a post-inversion filtering method
7) Compare the inversion to $R_{1.2\text{-}2.8}$ to check reliability of the inversion

In the following sections, we will first discuss the kernel function curves, followed by data pre-treatment – especially the criterion to retain and discard scans. Following that, an overview of the four inversion methods applied on three event types will be shown. These results will be inter-compared based on the sum of the inverted aerosol concentration and the aerosol size distributions in each size bin.


### 3 Results

### 3.1 Estimating the kernel function curves

The estimated kernel function curves, in short, are the derivative of the detection efficiency curves. Since the growth tube and the temperature of the saturator of the PSM do not change, a higher saturator flow rate will result in a higher supersaturation

ratio of DEG in the growth tube, resulting in higher detection efficiencies. This result is the opposite when the saturator flow rate is lower. The particle diameter cut-off sizes (taken at 50% detection efficiency) can be obtained from the calibration file provided by Airmodus Ltd., or by classifying charged particles with the use of a high flow Differential Mobility Analyzer. For further insight into obtaining the detection efficiency curves, refer to Cai et al. (2018). As the kernels are derived in laboratory conditions, it is important to note that the kernels may not be 1:1 accurate when measuring atmospheric particles due to the

unknown chemical composition, but nevertheless, they should be mathematically self-consistent, i.e., the corresponding detection efficiency increases with increasing particle diameter and saturator flow rate and the detection efficiency for large particles (e.g., 10 nm) does not vary in the used saturator flow rate range

### 3.2 Data retention rate and pre-inversion treatment

Table 1 presents the $p$-value statistical significance test to specific event days, with the total number of daily scans (from 00:00 to 23:59) and the number of retained and discarded scans. The test showed that both non-event and NPF events had a retention

rate over 90%, while haze events revealed an 80% retention rate. Of the three specific non-event days chosen, two had a high retention rate, while one had as many discarded scans as a haze event. A typical example of a measured, four-minute scan can be seen in Fig. 1. Retained scans (Fig. 1a and b) revealed a good correlation between the saturator flow rate and the measured particle concentration. In addition, the Spearman's rank correlation coefficient ($\rho$) of each retained scan was also significant. This contrasted discarded scans (Fig. 1c and d), which showed insignificant $\rho$ with no correlation between the saturator flow rate and particle concentration.

High retention rates may indicate presence of sub-3 nm particles, while lower retention rates, such as during haze days, may indicate that less sub-3 nm particles are present. As the aim is to invert high quality scans, only the scans with no significant correlation or negative correlation between the measured concentration and the saturator flow rate are discarded and, hence, the retention rate is around 80% even during haze days. However, the presence of sub-3 nm particles even in the presence of a high condensation sink also contribute to this high ratio of 80%, as discussed further in this study. Fig. 1c and 1d also present a typical challenge where the time resolution of the instrument (4 min. in this case) is larger than the time scale of the variations in the measured aerosol. High variations in the number concentration during one scan makes it oftentimes difficult to reliably invert data from a cumulative instrument, and indeed, the presented retention criteria may discard a large proportion of the scans that are mathematically difficult to invert.

### 3.3 Scan inversion

Individual, four-minute scans of both retained and discarded scans were inverted using the four inversion methods to assess the quality of the inversion and of the scan itself (Fig. 1). A measurable difference between retained and discarded inverted scans was observed. The inverted, discarded scans revealed concentrations close to zero for each size bin of each method, while inverted, retained scans revealed a quantifiable size distribution. From these inversions, one can make few observations. For example, all inversion methods give a rather similar-looking inverted size distribution for the retained scans, which suggests that all methods result in a reasonable inversion if the obtained raw data is good, which is in line with previous laboratory measurements by Cai et al. (2018). Certainly, the data quality check prior to data pre-treatment would ensure that the raw data considered good is retained while filtering out the bad data. The exception is with the step-wise method, which is sensitive to the slight air mass fluctuations that may lead to negative inverted or erroneous concentrations in some size bins. In the selected examples of the discarded scans (Fig. 1c and 1d), the kernel and step-wise methods' inverted concentrations yielded a particle size distribution despite the scan suggesting to not have any signals from sub-3 nm particles. As indicated above, the observed concentration fluctuations may have originated from air mass fluctuations. This means that with the measurement uncertainties, the use of the step-wise and kernel methods without prior data checking may lead to the inversion of artificial particle concentrations that are only revealed during the inversion. On the other hand, the H&A and EM methods appear to be much more robust against noisy data; after the inversions, these methods yielded no concentration at all from the

discarded scans. The significant differences in the behaviour of these inversion methods revealed measurement uncertainties that agree with the findings based on the Monte Carlo simulation in Cai et al. (2018).

### 3.4 Comparison of inversions to $R_{1.2-2.8}$ and total concentration

The inverted dataset was compared with $R_{1.2-2.8}$ of the same size range to estimate how well the inverted data is represented
(Fig. 2). The variable $R_{1.2-2.8}$ is calculated as the particle concentration difference between saturator flow rate at 1.3 lpm and 0.1 lpm. The sub-3 nm particle concentration estimate based on $R_{1.2-2.8}$ was more reliable the larger the sub-3 nm particle concentration was relative to the background particle concentration (i.e., all the particles outside the PSM sizing range, which in this study, $n > 2.8$ nm). If the ratio is low, the sub-3 nm particle concentration signal might not be distinguishable from the fluctuations of the background concentration. Further, since there are no corrections in the $R_{1.2-2.8}$ concentration estimate (e.g.,
losses), it might underestimate the real sub-3 nm particle concentration. During NPF events the step-wise inversion method reported the highest concentrations, about a factor of two larger than the kernel method, which showed the lowest concentrations. The H&A and EM methods reported concentrations were between the step-wise and the kernel method and closely resembled the concentration obtained from $R_{1.2-2.8}$. During non-NPF and haze periods, such that when $R_{1.2-2.8}$ was very noisy, the kernel method revealed clearly the highest concentrations, which are likely due to overestimation. This was already
observed in Fig. 1, where the kernel method-inverted concentrations were clearly inversion artefacts from scans that were discarded based on the insignificant correlation between the saturator flow rate and the measured particle concentration. The step-wise, EM and H&A methods showed rather similar concentrations, which were quite close to the values obtained from $R_{1.2-2.8}$ estimates. An interesting observation was during NPF events, where the H&A and EM methods before and after the NPF revealed none to very little concentration, which is contrary to what the kernel and step-wise methods and $R_{1.2-2.8}$ estimates
reported. This also could be observed during haze and non-event periods, but the differences are more subtle compared with the other inversion methods and $R_{1.2-2.8}$.

To get further insight into the overall performance of the inversions when the sub-3 nm particle concentration is low, histogram plots were made of the integrated concentration from the whole size range that is normalized with the total concentration
measured at the cut-off of 2.8 nm (0.1 lpm) (Fig. 3). Some observations can be made: the step-wise method is not sensitive to the concentrations because it is direct subtraction of concentrations; thus this may yield negative concentrations. The H&A and EM methods report high frequency at ~0, but also elevated frequencies below ratios less than 0.015 (or in absolute scale, less than about 200 cm$^{-3}$). This is in line with Cai et al. (2018), who show that the H&A and EM methods tend to report a near-zero size distribution when the sub-3 nm particle concentration is noisy and low compared to the background aerosol
concentration. In contrast, the kernel method never revealed ratios smaller than 0.015, which can be explained according to Figs. 1c and 1d – even when inverting data that clearly does not contain a signal from sub-3 nm particles, the kernel method nevertheless inverted some artificial particle concentrations. These artificial inverted concentrations originate from the random noise in the data that the inversion methods interpret as a real signal.

### 3.5 Overview of inversion for different types of events

### 3.5.1 New particle formation events

All NPF events in the study showed a typical increase of particle concentration with the highest concentrations observed around noon and minimally during the night (Fig. 4). All the methods revealed that the largest concentrations were observed in the smallest size bin. The EM, H&A and kernel methods revealed high concentrations in the largest size bin. Both EM and H&A methods showed very similar concentrations to one another. In contrast to the EM and H&A methods, the kernel and step-wise methods revealed a larger total concentration outside of the NPF event and the concentration intensity revealed no identifiable pattern. As discussed above, the difference is mainly caused by the behaviour of these inversion methods at a low signal-to-noise ratio.

### 3.5.2 Non-events

During non-event days, there were no indication of NPF events (Fig. 4). The distribution of the EM and H&A methods looked similar to each other in comparison with the kernel and step-wise methods. In addition, the H&A and EM methods revealed no particle sizes larger than 2 nm between 00:00–06:00 and 12:00–18:00, which largely contrasted the kernel and step-wise inversion methods. The step-wise method revealed scattered gaps with zero particle concentration in the size bins covered by the PSM throughout the day. This may be due to the limitation of the step-wise inversion algorithm. Since the algorithm is calculated as the difference between two adjoining size bins, if the difference is revealed as negative, the inversion itself would then have a gap in the size distribution. These gaps are more evident during noisier periods, such as during haze and non-event types. In contrast, the size distribution are latently smoothed in the H&A and EM methods.

### 3.5.3 Haze events

Similar to non-event days, the kernel and step-wise methods revealed particle concentrations in all size ranges throughout the day, while the H&A and EM methods showed concentrations predominantly in the lower size range. This led to the latter two methods being more qualitatively discernible compared with the kernel and step-wise methods. As with the other events, the EM method had large concentrations of particles at the highest size bin.

### 3.6 Comparison of inversion size bins

To compare the single size bins of each inversion method, four size bins were selected: 1.2-1.3 nm, 1.5-1.7 nm, 1.7-2.0 nm and 2.0-2.5 nm (Fig. 5). Three days were chosen to represent NPF, non-event and haze days (see also Figs. 2 and 4). On the NPF event, all the four inversion methods captured the diurnal trend of particle size distribution initiated by NPF. Considering measurement uncertainties, the inverted size distribution from different inversion methods generally agreed well with each

other, although the kernel method reported much lower overall size distribution. As seen from Fig. 2, the kernel method inversion clearly underestimated the NPF event particle size distribution and is also revealed in Fig. 5 in all but the largest size bin. Meanwhile, the step-wise method reported higher aerosol size distributions compared with the H&A and EM methods. Although the particle concentrations in the 1.2-1.3 nm channel was very similar, the difference in measured concentration was attributed to other sizes – particularly between the sizes 1.5-2.0 nm. The largest size bin (2.0-2.5 nm) revealed an interesting observation, such that both EM and H&A had lower concentrations compared with the step-wise and kernel methods. The latter two methods on 30 Jan showed a small peak at 07:00, which would be the approximate time that the NPF event begins (as seen in Fig. 2). It should be clarified that the true kernel functions are not determined due to the unknown aerosol chemical compositions. Hence, the differences between the inversion results may sometimes reflect the uncertainty of the measurement itself rather than simply quantify the difference of inversion methods.

On non-event and haze periods, newly formed clusters and particles are scavenged in a short period of time under the high coagulation sink in urban Beijing and their concentrations are presumably low (Cai et al., 2019). The EM method reported near-zero concentrations above 1.7 nm because it tends to report near-zero values when the particle concentration is low and noisy, as discussed in section 3.3. In contrast, the step-wise and kernel methods reported constantly existing concentration for particles larger than 1.7 nm. The similar phenomenon was also observed during the midnight of the NPF event. The methodology biases, e.g., the infinite-resolution assumption of the step-wise method and the instability of the least square method used in the kernel and H&A methods are the major causes of the background. Although the methods each revealed inversion challenges with measured atmospheric data, it is important to note that in chamber studies (e.g., Cai et al. (2018)), the inversion methods were rather robust.

## 4 Summary

In this study, we assessed the performance of four inversion methods: the step-wise, kernel, H&A and EM methods to invert PSM data measuring in real, atmospheric conditions. In addition, the study presented a novel method to pre-treat the data prior to inversion. The presented data employed a pre-treatment filter that scans the measured data to calculate the correlation between the observed particle concentration and supersaturation of a single scan. From the correlation analysis, scans are discarded when there is a significant non-correlation or negative correlation. The performance of the inversion methods were assessed by inverting single scans. All the methods were found to perform relatively well for scans that were measured during new particle formation events, while the inverted size distributions were overestimated with the kernel method when the data is noisy (i.e., during non-event and haze periods), and negative values can be obtained with the step-wise method when inverting noisy data. The EM and H&A methods were more robust when inverting noisy data, which in these cases, reported zeros. Since the variations in the background particle concentrations affected the performance of the inversion methods, one should be cautious when using any of these methods to approximate a size distribution when the total measured concentration and signal-to-noise ratio are low (in our study, less than ~500/cm$^3$ and ~0.02, respectively).

Based on the analysis presented in this study, there are many considerations that the user must be aware when inverting PSM data:

425      1)   When inverting PSM data, a good guideline to follow is by first creating a similar workflow as this study (see section 2.6). Ideally, users should stop between each step to examine the data output to ensure that it looks reasonable before continuing. This way, users will know where, during the inversion process, the problem lies.

     2)   Selection of the size channels for inversion is important and largely depends on the instrument-specific calibration. First, the channels need to fall within the calibration curve limits and that each size diameter limit has its own distinct saturator flow. The latter is especially important because at diameters greater than 2 nm, the saturator flow versus diameter (calibration) curve begins to flatten out.

     3)   Data pre-treatment is an important part of the inversion to obtain reliable data. Scans that contain a clearly unphysical correlation between the measured concentration and supersaturation should be discarded. An unphysical correlation is one where the PSM's saturator flow rate is not positively correlated with the measured total concentration. We employ a Spearman's rank correlation coefficient with the significance set at $p < 0.05$ to ensure data is of high quality. Naturally, changing this significance threshold would yield stricter or relaxed restrictions, likely resulting in fewer or more retained scans, respectively.

     4)   In this manuscript, we used four-minute scans (i.e., the combination of an upward and downward scan of the saturator flow) as the time resolution of our measurements. Alternatively, scans can be selected at a higher two-minute resolution or at lower resolutions (i.e., > four-minute scan). The selection of the scanning length is a compromise between better quality data and higher time resolution. In our case study, we measured urban atmospheric particles where growth rates at this size range can be approximately 1 nm/hr. Therefore, selecting four-minute scans is a reasonable time resolution.

     5)   It is strongly advised to invert the data with more than one inversion method to compare the results with one another, rather than accepting blindly the inverted values of one method. A comparison would affirm whether the inverted measured concentration is real and they are in good agreement (e.g., within a factor of 0.5-2; see point 7, below). If the comparison does not agree then the user should check that the kernel function curves are reasonable (see Cai et al. (2018)).

     6)   The recommended method to retrieve the particle size distribution of PSM data is the EM method. From this study, the EM and H&A performed similarly, however based on theoretical understanding (see Cai et al. (2018)), the EM method is more stable of the two (i.e., the inversion is more smooth and the concentration is more continuous). The kernel method should not be used to invert PSM data during a non-NPF event and used cautiously during NPF events. This is because inversions may be over- or under-estimated, and at worst, artificial counts can be created by the inversion itself.

7)   The measured size distributions of ambient aerosols should be reported using a limited number of size bins (e.g., 4- to 6-channels) because the assumed inversion kernels may deviate from the true kernels.

8)   To improve data reliability, comparability and availability, the used inversion method and the measured size distribution functions ($dN/dlogD_p$ vs $D_p$) should be reported together with any other subsequent analysis from the PSM data.

9)   As a first approximation, the PSM user should compare the inverted total sub-3 nm particle concentration to the sub-3 nm concentration obtained from the raw data by subtracting the concentration measured at lowest supersaturation from the concentration measured with highest supersaturation ($R_{1.2-2.8}$). These concentrations should be comparable (within a factor of 0.5-2). However, if the inverted data does not correspond well with $R_{1.2-2.8}$, check with other inversion methods – the deviation may be due to the inversion or bad data quality.

10)   A signal-to-noise ratio test can be performed to determine what the smallest concentration can be detected with the PSM. This would help users identify the measurement limits of the instrument from the data. In our study, we found that the site-specific ratio was approximately 0.02. Nevertheless, as a safety limit, we advise users to use data 2-3 times higher than their calculated ratio.

11)   Most importantly, the performance of the PSM should be checked regularly and the detection efficiency (that
470         determines the inversion kernel) should be calibrated sporadically because the kernel information is used in the EM, H&A and kernel inversions.

12)   The MATLAB code written for this study is available via GitHub (https://github.com/tommychan-dev/PSM-Inversion). Sample atmospheric data with the PSM calibration file can also be found there.


**Figure Captions:**

**Fig. 1**. Four-minute retained (a, b) and discarded (c, d) scans (left column) using the PSM and their inversion into a particle size distribution (right column). The black dots indicate the (two-minute) upward scan of the saturator flow; and the blue dots indicate the corresponding (two-minute) downward scan of the saturator flow. The red line indicates the regression line of the four-minute scan. An 11-channel size bin range was used for the inversion.

**Fig. 2**. Comparison between inverted and raw data (i.e, $R_{1.2\text{-}2.8}$; calculated as the difference between saturator flow rate at 1.3 lpm and 0.1 lpm). Event types shown are NPF (left; 30 Jan. 2018), non-event (center; 6 Mar. 2018) and haze (right; 9 Mar. 2018).

**Fig. 3**. Signal-to-noise ratio of all selected study days. Data shown is the integrated concentration from the whole size range with the total concentration measured at the cut-off of 2.8 nm (0.1 lpm).

**Fig. 4**. Step-wise, kernel, H&A and EM inversions of a selected NPF (left; 30 Jan. 2018), non-event (center; 6 Mar. 2018) and haze (right; 9 Mar. 2018) event.

**Fig. 5**. Comparison of size bins from inversions. Event types shown are NPF (left; 30 Jan. 2018), non-event (center; 6 Mar. 2018) and haze (right; 9 Mar. 2018).

**Table 1**. List of selected days for analysis with their event type, number of scans and retention rate. All data are measured in 2018.

| Event | Date (m/d) | No. of scans | Retained | Discarded |
|---|---|---|---|---|
| NPF | 30 Jan | 362 | 350 | 12 |
| | 31 Jan | 359 | 336 | 23 |
| | 09 Feb | 353 | 326 | 27 |
| | 16 Feb | 357 | 321 | 36 |
| Non-event | 01 Feb | 361 | 340 | 21 |
| | 13 Feb | 361 | 296 | 65 |
| | 23 Feb | 361 | 342 | 19 |
| | 06 Mar | 361 | 283 | 78 |
| Haze | 17 Feb | 361 | 285 | 76 |
| | 26 Feb | 361 | 301 | 60 |
| | 02 Mar | 361 | 283 | 78 |
| | 09 Mar | 361 | 282 | 79 |


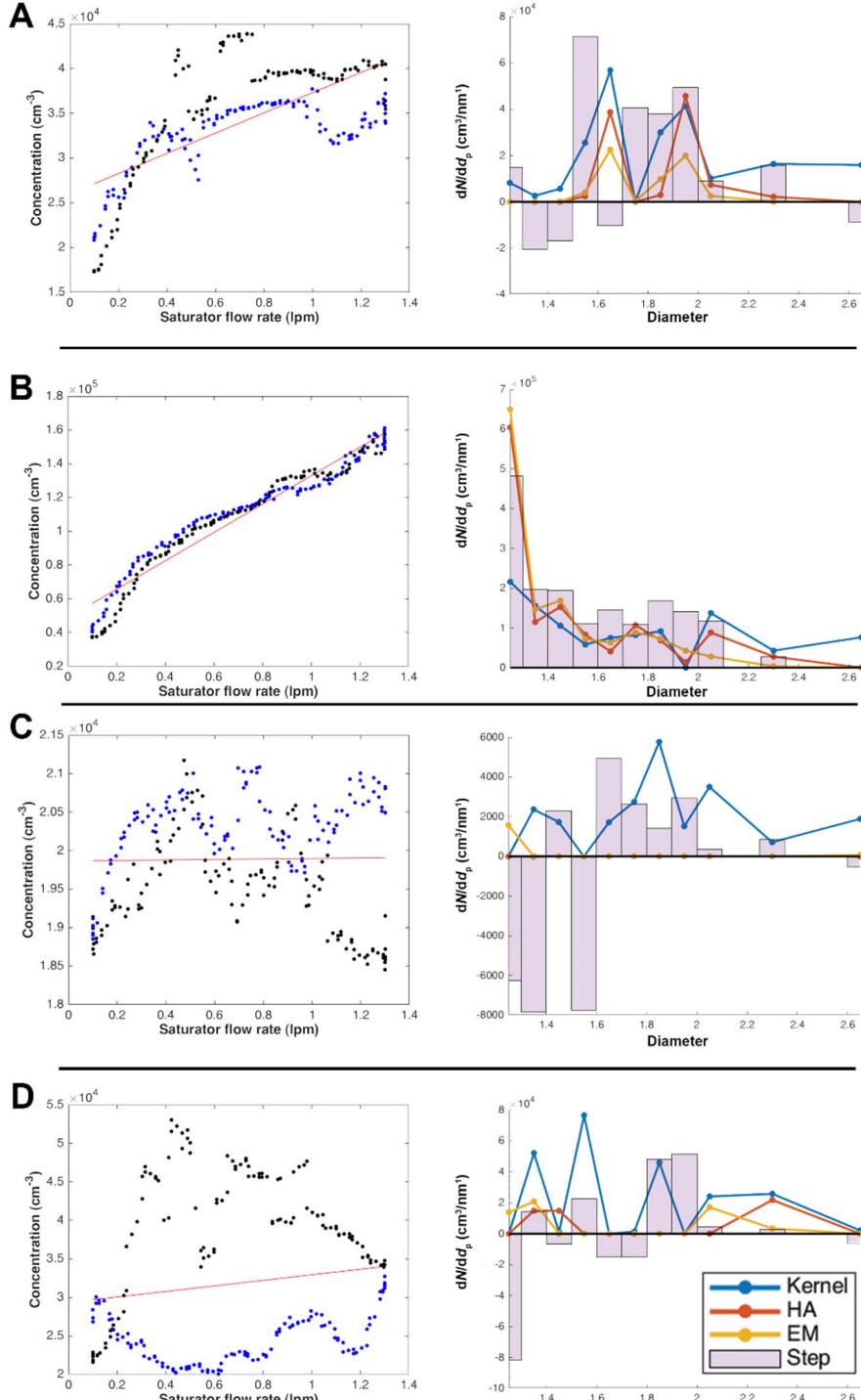

**Fig. 1**


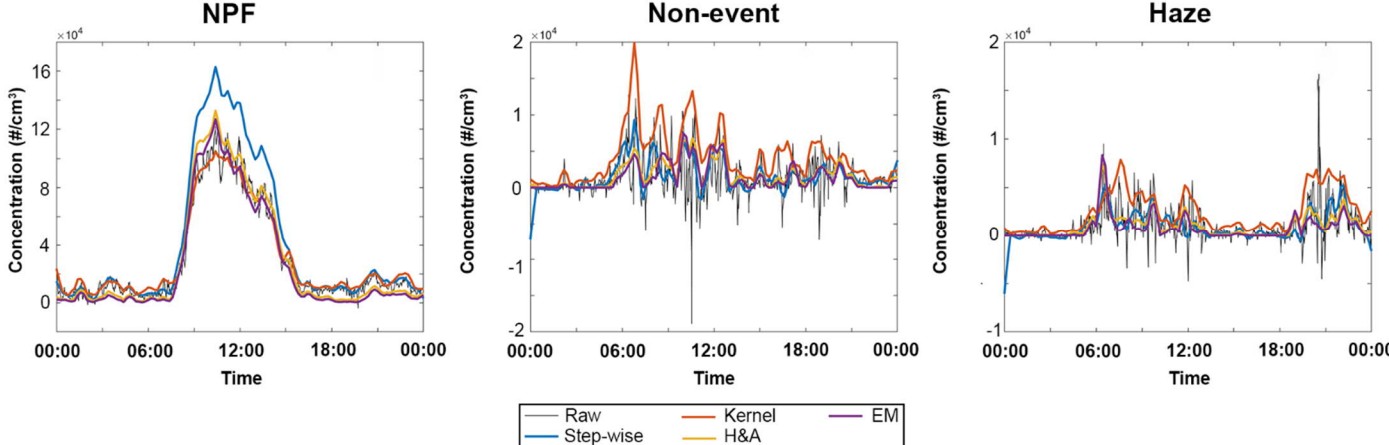

**Fig. 2**.

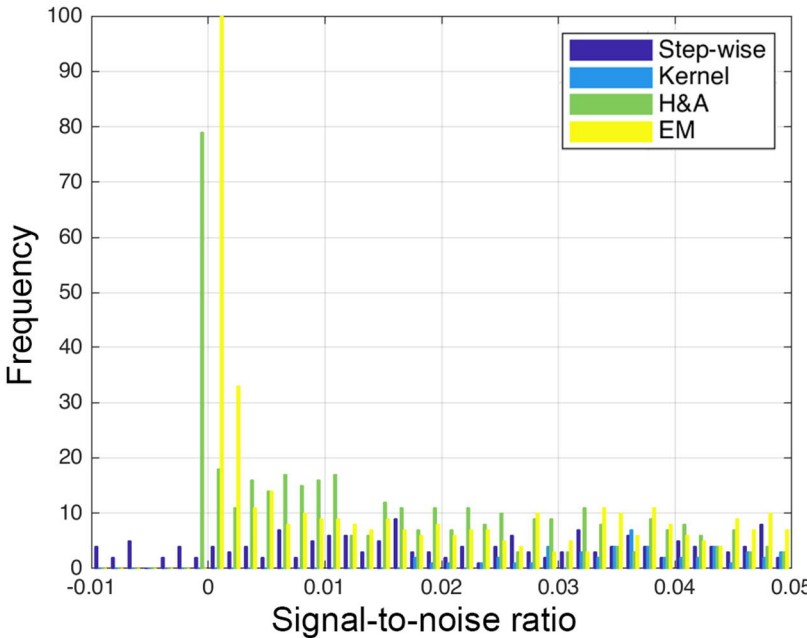

**Fig. 3**.


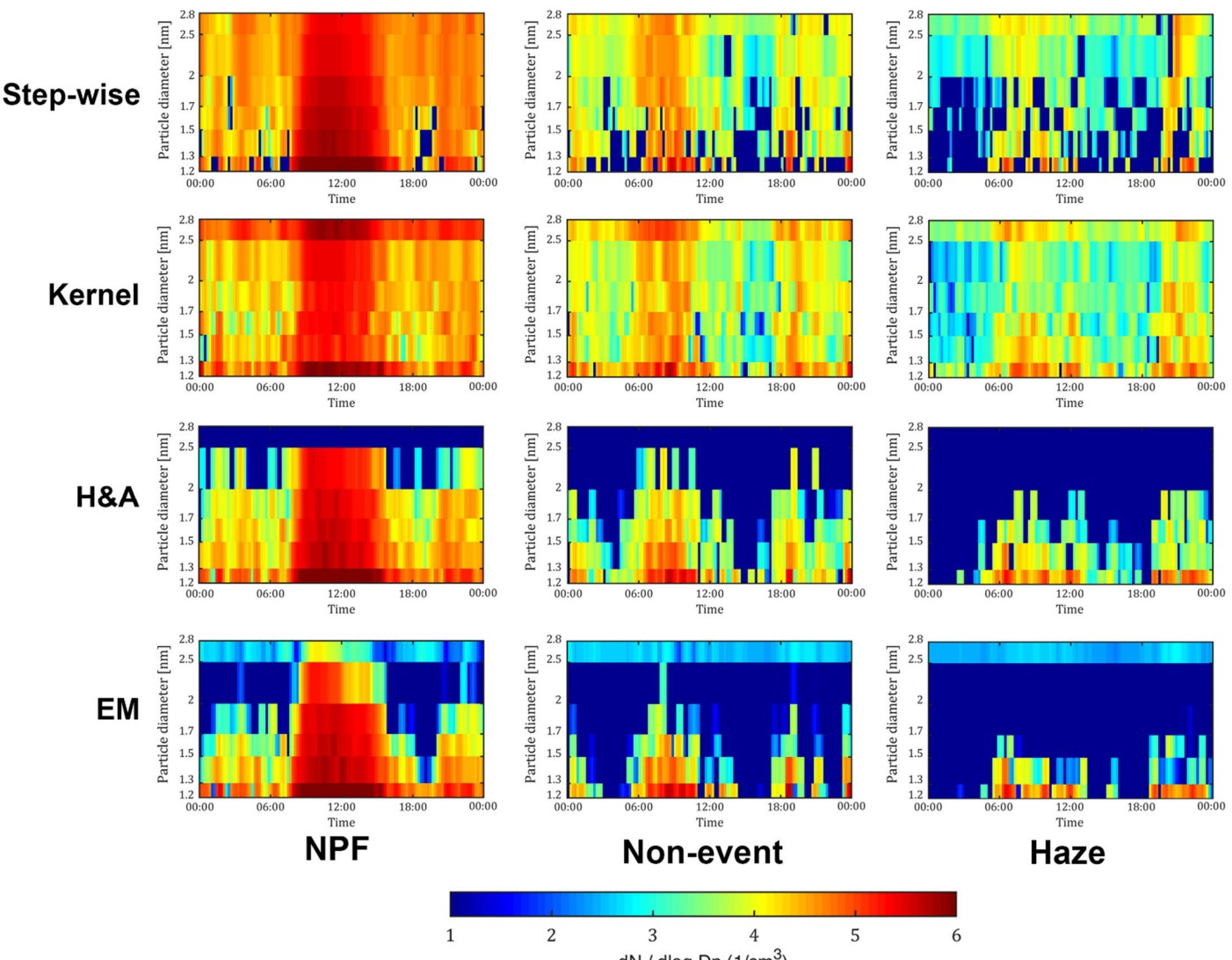

Fig. 4.


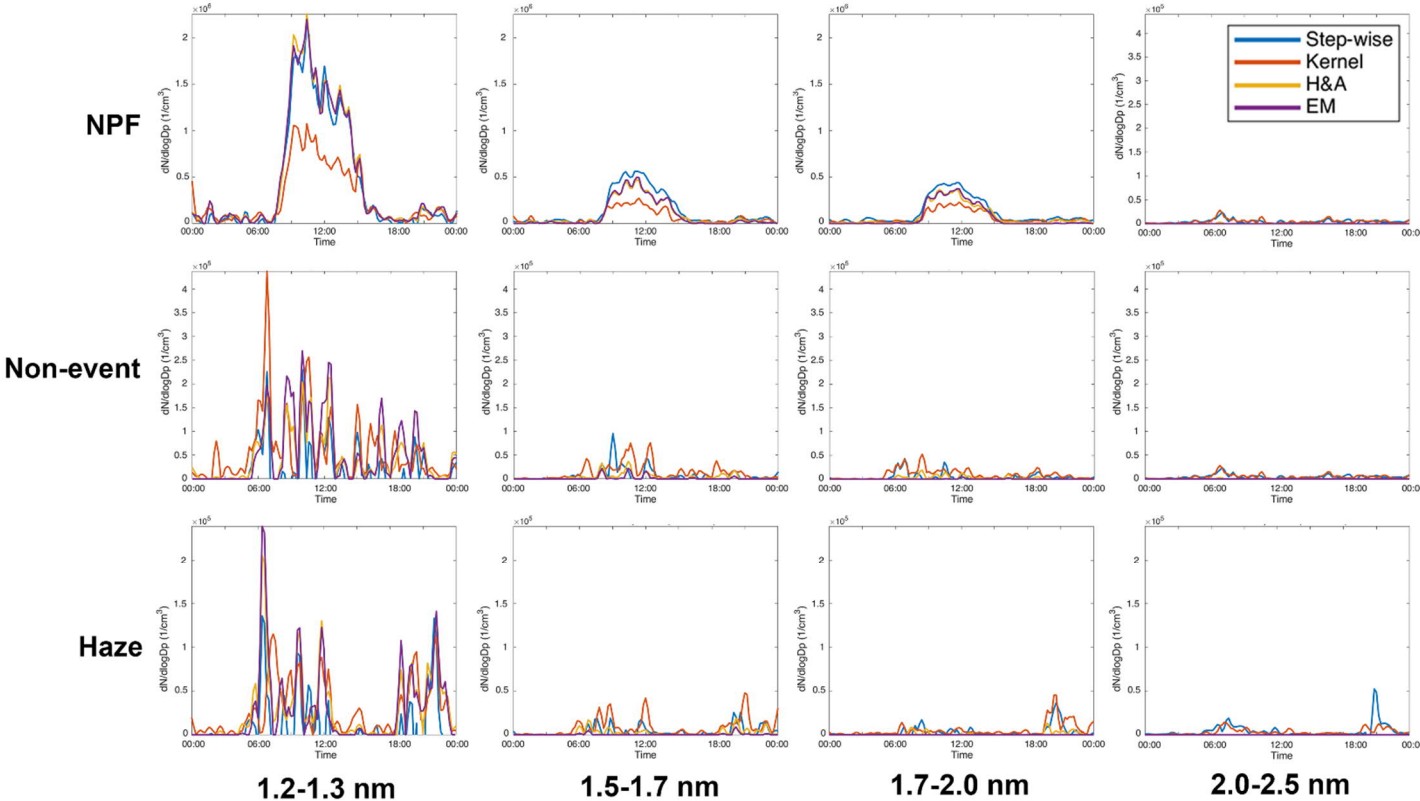

**Fig. 5**.


## 5 Code availability

MATLAB inversion codes used for this study are available via GitHub (https://github.com/tommychan-dev/PSM-Inversion).

## 6 Author contribution

TC, RC and JK designed the study and TC carried them out. TC, RC, LA, JV and LD developed the inversion code. YL
(Fudan), LW, YZ, YC, YL (BUCT) and MK provided the facilities, instruments and funding for the study. TC prepared the manuscript with contributions from all co-authors.

## 7 Competing interests

The authors declare that they have no conflict of interest.

## 8 Acknowledgements

We wish to thank Rima Baalbaki for her insightful comments and code debugging prowess. This work was supported by the University of Helsinki, Faculty of Science support grant (#75284140), 3-year grant (#75284132) and the Finnish Academy of Science project (#1325656).

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
