# Peer review of "Assessment of particle size magnifier inversion methods to obtain particle size distribution from atmospheric measurements"

_Atmospheric Measurement Techniques, 2019_

## Referee Comment (RC1) · Anonymous Referee #1 · 19 Feb 2020

The paper by Chan et al. discusses the applicability of four different inversion methods for data measured with a particle size magnifier (PSM). The PSM is a condensation particle counter that measures the particle number density for particles as small as ~1 nm. By adjusting the saturator flow rate and thereby the supersaturation, different activation diameters can be adjusted. Scanning of the saturator flow rate allows changing the activation size between ca. 1 and 3 nm. In order to retrieve a size distribution for this diameter range an inversion algorithm needs to be applied to the raw data. Chan et al. test the performance of four different inversion algorithms for raw data measured in Beijing between January and March 2018. The conclusion is that one inversion method (EM method) is the preferred one compared to the other ones. In principle,

the presented study is helpful for the community working with PSM data. However, the recommendations provided in Section 4 of the paper are currently insufficient. The authors provide a list of recommendations without defining clear criteria for the rejection of data and the assessment of the accuracy of the inversion. In this respect, the paper should be more specific and needs to be revised before publication.

General comments

P4, L116/L117: The authors should add a comment in how far line losses could affect the outcome of the inverted size distribution. Or are the line losses not at all relevant?

P6, L166: Shouldn't the unit be particles cm-3 nm-1 (as in equation (1) n is multiplied with ddp (unit nm))?

P6, L166: please specify the source(s) of the errors e_i

P6, equation (2): it is not clear why s_max is used in this equation instead of s_i; please add a comment and explain why this is the case

Fig. 1 (and discussion starting P9, L257): It seems that for the inversions always the combined data of one up and one down scan are used. Can the authors please explain why this is done? In order not to lose information for rapidly changing conditions it could make sense to treat the up and down scans separately.

Fig. 1: Please use different colors for the data corresponding to an up or down scan, respectively.

Fig. 1: It is very hard to read this figure (this applies to all figures) due to a very small size. Additionally, it is not clear how the size bins at the right hand side were chosen. In the text it says the size distribution ranged from 1.2 to 2.8 nm with 6 size channels (P7, L215/216). However, in Fig. 1 many more channels are visible and the size distribution ends at ∼2.6 nm.

P9, L278: What do the authors mean by "data is good"? Please be more specific in

terms of what distinguishes "good" from "bad" data.

P10, L316: Please add some discussion why generally no constrain is applied that forces the concentrations in each size bin to be non-negative. Furthermore, the authors should add some discussion why no normalization with respect to the total number concentration (difference between the concentrations in the smallest and largest size bin is performed). This would avoid over- or underestimation of the concentrations.

P13, L393: Please define how the signal to noise ratio can be estimated for the PSM data.

P13, L398/399: A mathematical criterion should be provided that allows to identify such an unphysical correlation.

P13, L400 to 402: What should be done if that comparison yields a difference? How much of a difference should be tolerable? Why is the inverted data not constrained to match the measured concentration difference?

P13, L414: The authors should explain why the EM method is the preferred one. From the previous discussions it seemed that both the H&M and the EM method give similar results.

P13, L415: It is not clear what follows from such a comparison and how it should be performed. What is the criterion for good or bad agreement and what should be done if the agreement is not good?

Minor comments

P1, L18: "haze event"

P2, L41: "one" instead of "once"

P2, L50: "the activated particles are further grown"

P2, L51: "are activated in the first stage"

P3, L97: "5.33 mm/month"?

P4, L11: "can measure a maximum particle concentration of up to 10ˆ5 #/cm3"

P4, L114: "so that they can"

P4, L127: delete the word "be"

P4, L131: "the data were"

P6, L165: "where Ri is the particle number concentration for a saturator flow rate of si"

P7, L205: "directly"

P9, L275: "little to no particle size concentration", please reformulate, e.g., to "concentrations close to zero for each size bins"

P9, 278: "This is in line with"

P9, 286: "these methods inverted very little to no concentrations at all from discarded scans"; please reformulate, e. g., to "after the inversions these methods yielded concentrations"

P10, L313: "who" instead of "which"

P11, L335: What do the authors mean by "large"?

P11, L336: "with no particle concentration"; do the authors mean "with zero particle concentration in the size bins covered by the PSM"?

P13, L392: "approximate" instead of "appropriate"?

P16, L436: "Comparison between inverted and raw data"

---

## Referee Comment (RC2) · Anonymous Referee #2 · 31 Mar 2020

Chan et al. present a comparison of 4 different sub-3 nm inversion methods for the PSM. This comparison examines atmospheric observations in Beijing, China in contrast to previous studies which focused on laboratory measurements. Their results show that low raw counts led to unrealistic size distributions from step-wise and kernel methods. Overall, this paper goes into detail the four inversion methods. Their conclusions that the higher raw counts lead to more accurate data inversion and better agreement between the methods. This is not super surprising. Thus, the authors should instead focus more of their interesting results on their pre-treatment of the data. The manuscript is well written however it is not clear from their observations what the community should do. The authors need to address the below major comments before

this paper can be consider for publication.

Major comments: Line 271: The authors used rather long scan times or 4 minutes. During nucleation and growth events, particles are rapidly changing in concentration and diameter (up to 100 nm/hr growth rates have been observed). Are these analysis techniques and inversion methods going to be useful in these situations?

Line 308: It would be helpful if the authors could put signal to noise bounds on when each inversion method works/doesn't work. In this paragraph, they mention a ratio but it's not obvious how this relates to signal to noise which is a more used parameter. (It's also not explained well what this ratio is they are referring to.) The conclusions mention signal to noise of ~0.02. Where did this come from?

Line 343: The constant pool of 1.7 nm particles is attributed to coagulation controlled particle growth. This seems like a bold state to claim without more results. The authors should better justify this with coagulation calculations or composition measurements etc. However, I realize this is outside the scope of this study so may be best to just remove this paragraph.

All of the Figures are impossible to read. I could not determine if the science presented was sound because the figures are tiny and the font is blurred. The SI figures need to be fixed as well.

The most important issue of this study is that it is not clear what we the community should do. The authors tried to lay out specific steps but the message is confusing. Which inversion method should we use; is EM applicable for all conditions? What are the uncertainty bounds on the inversion methods? Is the PSM's proprietary inversion method (I didn't think a method could be proprietary) going to move to EM?

Minor comments:

Line 35: bit misleading to say SMPS measures sub 3 nm size distribution and then cite Wang and Flagan. More accurate to cite nano DMA and ultrafine CPC or DEG

CPC papers, or their combined SMPS from Jinkun Jiang. Also, why not include mass spectrometers?

Line 41: once should be one

Line 60: which manufacture? No where do the authors say Airmodus.

Line 124: I know this is outside this paper but I've always wondered about the mixing efficiency and wall loss changes as the saturator flow is varied from 0.1-1.3 LPM.

Line 135: How do the authors know there were periods of no sub 3 nm particles if they're relying on their inversion method to tell them this information? Their statements would be more justifiable if there were a second instrument measuring particles in addition to the PSM. The authors spend the whole paper getting the readers to think there is major uncertainty in the PSM measurement/inversion method then to say that they relied on it to determine when 3-nm particles were present seems a bit contradictory.

Section 2.5: The authors define $R_i$ to be different (particle number concentration or particle concentration raw counts) between each inversion method and it's very confusing. This get more confusing with the authors refer to R1.2-2.8 and total concentration. What $R_i$ is that?

Line 232: Are these diameters mobility diameters or geometric diameters?

---

## Author Comment (AC1) · 29 May 2020

–P4, L116/L117: The authors should add a comment in how far line losses could affect the outcome of the inverted size distribution. Or are the line losses not at all relevant?

-We have added a new sentence with references in regards to the line losses

–P6, L166: Shouldn't the unit be particles cm-3 nm-1 (as in equation (1) n is multiplied with ddp (unit nm))?

-You are correct, we have changed the units

–P6, L166: please specify the source(s) of the errors e_i

[Figure]

-We have added a sentence of some errors

–P6, equation (2): it is not clear why s_max is used in this equation instead of s_i; please add a comment and explain why this is the case

-We have included the derivation of the step-wise method in the SI section

–Fig. 1 (and discussion starting P9, L257): It seems that for the inversions always the combined data of one up and one down scan are used. Can the authors please explain why this is done? In order not to lose information for rapidly changing conditions it could make sense to treat the up and down scans separately.

-Combining two scans is a compromise between better data quality and better time resolution. Separating the scans yields better time resolution, but if the data is noisy, averaging two scans may yield better quality data as averaging two individual scans can average out random fluctuations. In our study, we measured the urban atmosphere where particle growth can be approx. 1nm/hr, we do not need such high resolution. However, chamber experiments or measuring car exhaust, doing separate scans may be ideal. We have expanded the explanation of the time resolution in the summary section

–Fig. 1: Please use different colors for the data corresponding to an up or down scan, respectively.

-We have added different colours to separate the up and down scan

–Fig. 1: It is very hard to read this figure (this applies to all figures) due to a very small size. Additionally, it is not clear how the size bins at the right hand side were chosen. In the text it says the size distribution ranged from 1.2 to 2.8 nm with 6 size channels (P7, L215/216). However, in Fig. 1 many more channels are visible and the size distribution ends at approx. 2.6 nm.

-You are correct, Fig. 1 used 11 channels for the inversion of the single scan. We have changed the text to reflect this and include the size channels used.

–P9, L278: What do the authors mean by "data is good"? Please be more specific in terms of what distinguishes "good" from "bad" data.

-We have added a sentence to indicate what is "good" data and "bad" data.

–P10, L316: Please add some discussion why generally no constrain is applied that forces the concentrations in each size bin to be non-negative. Furthermore, the authors should add some discussion why no normalization with respect to the total number concentration (difference between the concentrations in the smallest and largest size bin is performed). This would avoid over- or underestimation of the concentrations.

-This sentence did not correlate with our results and we have removed it. We did not force the concentrations in each size bin to be zero – rather the total concentration. We do not understand why we would normalize. What we see in Fig. 2 reveals already whether the inversion has been over- or underestimated.

–P13, L393: Please define how the signal to noise ratio can be estimated for the PSM data.

-We have added this in the summary section

–P13, L398/399: A mathematical criterion should be provided that allows to identify such an unphysical correlation.

-We have now provided an explanation of how the thresholds are calculated in the data quality check and in the summary section

–P13, L400 to 402: What should be done if that comparison yields a difference? How much of a difference should be tolerable? Why is the inverted data not constrained to match the measured concentration difference?

-We have added a few sentences to elaborate on this issue in the summary section.

–P13, L414: The authors should explain why the EM method is the preferred one. From the previous discussions it seemed that both the H&M and the EM method give similar
results.

-We have added a few sentences to explain the EM method and clarified the section to make it easier to understand for the reader

–P13, L415: It is not clear what follows from such a comparison and how it should be performed. What is the criterion for good or bad agreement and what should be done if the agreement is not good?

-We have clarified this in the summary section.

–P1, L18: "haze event"

-Changed

–P2, L41: "one" instead of "once"

-Changed

–P2, L50: "the activated particles are further grown"

-Changed

–P2, L51: "are activated in the first stage"

-Changed

–P3, L97: "5.33 mm/month"?

-It is the average of all three months. We added 'overall' to the sentence.

–P4, L11: "can measure a maximum particle concentration of up to 10e5 #/cm3"

-Changed

–P4, L114: "so that they can"

-Changed

–P4, L127: delete the word "be"

-Changed

–P4, L131: "the data were"

-Changed

–P6, L165: "where Ri is the particle number concentration for a saturator flow rate of si"

-Changed

–P7, L205: "directly"

-Changed

–P9, L275: "little to no particle size concentration", please reformulate, e.g., to "concentrations close to zero for each size bins"

-Changed

–P9, 278: "This is in line with"

-Changed

–P9, 286: "these methods inverted very little to no concentrations at all from discarded scans"; please reformulate, e. g., to "after the inversions these methods yielded concentrations"

-Changed

–P10, L313: "who" instead of "which"

-Changed

–P11, L335: What do the authors mean by "large"?

-Larger than 2 nm. We have changed the text.

–P11, L336: "with no particle concentration"; do the authors mean "with zero particle concentration in the size bins covered by the PSM"?

-This is correct and we have changed the text

–P13, L392: "approximate" instead of "appropriate"?

-Changed

–P16, L436: "Comparison between inverted and raw data"

-Changed

Please also note the supplement to this comment:
https://www.atmos-meas-tech-discuss.net/amt-2019-465/amt-2019-465-AC1-supplement.pdf

———————————————————

[Figure]

[Figure]

**Fig. 1.**

[Figure]

**Fig. 2.**

[Figure]

**Fig. 3.**

[Figure]

**Step-wise**

**Kernel**

**H&A**

**EM**

**NPF**  **Non-event**  **Haze**

dN / dlog Dp (1/cm$^3$)

**Fig. 4.**

**NPF**

**Non-event**

**Haze**

**1.2-1.3 nm**  **1.5-1.7 nm**  **1.7-2.0 nm**  **2.0-2.5 nm**

Legend: Step-wise, Kernel, H&A, EM

**Fig. 5.**

[Figure]

**Fig. 6.**

Fig. 7.

**Supplement:**

**Supplementary Information**

The derivation of the step-wise method is as follows:

$$\frac{\mathrm{d}R}{\mathrm{d}s}(s_{\mathrm{i}}) = \int_{0}^{+\infty} K(s_{\mathrm{i}}, d_{\mathrm{p}}) \times \frac{\mathrm{d}N}{\mathrm{d}d_{\mathrm{p}}} \mathrm{d}d_{\mathrm{p}}$$

$$s_{\mathrm{i}} \Leftrightarrow d_{\mathrm{i}} : \int_{0}^{+\infty} K(s_{\mathrm{i}}, d_{\mathrm{p}}) \times \mathrm{d}d_{\mathrm{p}} = \int_{d_{\mathrm{i}}-\delta}^{d_{\mathrm{i}}+\delta} K(s_{\mathrm{i}}, d_{\mathrm{p}}) \times \mathrm{d}d_{\mathrm{p}}, \delta \to 0$$

$$\therefore \frac{\mathrm{d}R}{\mathrm{d}d_{\mathrm{p}}}(d_{\mathrm{i}}) = \frac{\mathrm{d}R}{\mathrm{d}s}(s_{\mathrm{i}}) \times \frac{\mathrm{d}s}{\mathrm{d}d_{\mathrm{p}}} = \frac{\mathrm{d}s}{\mathrm{d}d_{\mathrm{p}}} \int_{0}^{+\infty} K(s_{\mathrm{i}}, d_{\mathrm{p}}) \times \frac{\mathrm{d}N}{\mathrm{d}d_{\mathrm{p}}} \mathrm{d}d_{\mathrm{p}}$$

$$= \int_{d_{\mathrm{i}}-\delta}^{d_{\mathrm{i}}+\delta} K(s_{\mathrm{i}}, d_{\mathrm{p}}) \times \frac{\mathrm{d}N}{\mathrm{d}d_{\mathrm{p}}} \times \frac{\mathrm{d}s}{\mathrm{d}d_{\mathrm{p}}} \times \mathrm{d}d_{\mathrm{p}}$$

$$= \frac{\mathrm{d}N}{\mathrm{d}d_{\mathrm{p}}}(d_{\mathrm{i}}) \int_{s_{\mathrm{i}}-\varepsilon}^{s_{\mathrm{i}}+\varepsilon} K(s_{\mathrm{i}}, d_{\mathrm{i}}) \times \mathrm{d}s$$

$$= \frac{\mathrm{d}N}{\mathrm{d}d_{\mathrm{p}}}(d_{\mathrm{i}}) \int_{0}^{+\infty} K(s_{\mathrm{i}}, d_{\mathrm{i}}) \times \mathrm{d}s$$

$$= \frac{\mathrm{d}N}{\mathrm{d}d_{\mathrm{p}}}(d_{\mathrm{i}}) \times \eta(d_{\mathrm{i}}, s_{\max}),$$

where d is the derivate symbol; $R$ is the raw concentration; $s$ is the saturator flow rate; $K$ is the kernel; $d\mathrm{p}$ is the particle diameter; $s_{\mathrm{i}}$ and $d_{\mathrm{i}}$ are the saturator flow rate and diameter at the *i*th point, respectively; $\delta$ is Cauchy's definition of the limit; $\varepsilon$ is the error; $s_{\max}$ is the maximum saturator flow rate; $\eta$ is the overall detection efficiency; and $\int$ is the integral symbol. In the stepwise method, the resolution of the kernel is assumed to be positive infinity and hence, there is a one-to-one relationship between each saturator flow rate and the retrieved particle diameter.

[Figure]

Fig. S1. Comparison of the 6- (dotted) and 11-channel (solid) size bins. The grey line indicates the raw estimation, $R_{1.2\text{-}2.8}$ (calculated as difference between saturator flow rate at 1.3 lpm and 0.1 lpm). Top figures are during an NPF event (30 Jan.) and bottom figures are from a non-event day (6 Mar.).

[Figure]

Fig. S2. Inversion kernel using 6-channels. Note that these kernels were theoretically calculated and not implicitly used for the current study.

---

## Author Comment (AC2) · 29 May 2020

-Line 271: The authors used rather long scan times or 4 minutes. During nucleation and growth events, particles are rapidly changing in concentration and diameter (up to 100 nm/hr growth rates have been observed). Are these analysis techniques and inversion methods going to be useful in these situations?

-As mentioned earlier to Referee 1's comment, we can expect particle growth to be

For a PSM, a 4-minute scan results in a time resolution of upwards to 15 nm/hr

-Line 308: It would be helpful if the authors could put signal to noise bounds on when each inversion method works/doesn't work. In this paragraph, they mention a ratio but it's not obvious how this relates to signal to noise which is a more used parameter. (It's also not explained well what this ratio is they are referring to.) The conclusions mention signal to noise of âLij0.02. Where did this come from?

-We have now added in the summary section a more detailed explanation of the signalto-noise ratio calculation and how users can use this.

-Line 343: The constant pool of 1.7 nm particles is attributed to coagulation controlled particle growth. This seems like a bold state to claim without more results. The authors should better justify this with coagulation calculations or composition measurements etc. However, I realize this is outside the scope of this study so may be best to just remove this paragraph.

-We have removed the paragraph

-All of the Figures are impossible to read. I could not determine if the science presented was sound because the figures are tiny and the font is blurred. The SI figures need to be fixed as well.

-We have made all the figures more legible

-The most important issue of this study is that it is not clear what we the community should do. The authors tried to lay out specific steps but the message is confusing. Which inversion method should we use; is EM applicable for all conditions? What are the uncertainty bounds on the inversion methods? Is the PSM's proprietary inversion method (I didn't think a method could be proprietary) going to move to EM?

-We have expanded the summary section to thoroughly detail recommendations/user considerations. We do not know Airmodus' future in adopting a proprietary method. Here, we simply report our analysis. In regards to the uncertainty bounds, we did not

AMTD
discuss the inversion specific uncertainties. However, we provide the code and sample data used in this study so the user may estimate the uncertainties in the PSM data.

-Line 35: bit misleading to say SMPS measures sub 3 nm size distribution and then cite Wang and Flagan. More accurate to cite nano DMA and ultrafine CPC or DEG CPC papers, or their combined SMPS from Jinkun Jiang. Also, why not include mass spectrometers?

-We have removed the reference and cited properly. Mass specs are out of the scope of this study.

-Line 41: once should be one

-Changed

-Line 60: which manufacture? No where do the authors say Airmodus.

-We have added Airmodus to the sentence and also in the Introduction, when it was first mentioned

-Line 124: I know this is outside this paper but I've always wondered about the mixing efiňĄciency and wall loss changes as the saturator flow is varied from 0.1-1.3 LPM.

-We are not sure if there are any studies regarding this, but theoretically, there are changes. However, these will not affect the inversion methods. In addition, the instrument function is always calibrated in the lab and it includes all changes in the mixing efficiency and any losses.

-Line 135: How do the authors know there were periods of no sub 3 nm particles if they're relying on their inversion method to tell them this information? Their statements would be more justifiable if there were a second instrument measuring particles in addition to the PSM. The authors spend the whole paper getting the readers to think there is major uncertainty in the PSM measurement/inversion method then to say that they relied on it to determine when 3-nm particles were present seems a bit contradictory.
-We have modified the sentence to make it clearer. In addition, we determine whether there are no sub-3 nm particles based on the correlation between saturator flow rate and concentration. It is not physically possible if there is a negative correlation. In addition, when concentration of sub-3 nm particles is high, it is reasonable to believe that the particles are there (Kangasluoma 2020). At low concentrations close to the detection limit of the PSM, problems may arise and we attempt to discuss this in the manuscript.

-Section 2.5: The authors define Ri to be different (particle number concentration or particle concentration raw counts) between each inversion method and it's very confusing. This get more confusing with the authors refer to R1.2-2.8 and total concentration. What Ri is that?

-This is a mis-type. R\_i is the raw concentration and we have changed the text to reflect this.

-Line 232: Are these diameters mobility diameters or geometric diameters?

-Mobility. We have modified the sentence

Please also note the supplement to this comment: https://www.atmos-meas-tech-discuss.net/amt-2019-465/amt-2019-465-AC2supplement.pdf
Fig. 1.

**Supplement:**

**Supplementary Information**

The derivation of the step-wise method is as follows:

$$\frac{\mathrm{d}R}{\mathrm{d}s}(s_{\mathrm{i}}) = \int_{0}^{+\infty} K(s_{\mathrm{i}}, d_{\mathrm{p}}) \times \frac{\mathrm{d}N}{\mathrm{d}d_{\mathrm{p}}} \mathrm{d}d_{\mathrm{p}}$$

$$s_{\mathrm{i}} \Leftrightarrow d_{\mathrm{i}} : \int_{0}^{+\infty} K(s_{\mathrm{i}}, d_{\mathrm{p}}) \times \mathrm{d}d_{\mathrm{p}} = \int_{d_{\mathrm{i}}-\delta}^{d_{\mathrm{i}}+\delta} K(s_{\mathrm{i}}, d_{\mathrm{p}}) \times \mathrm{d}d_{\mathrm{p}}, \delta \to 0$$

$$\therefore \frac{\mathrm{d}R}{\mathrm{d}d_{\mathrm{p}}}(d_{\mathrm{i}}) = \frac{\mathrm{d}R}{\mathrm{d}s}(s_{\mathrm{i}}) \times \frac{\mathrm{d}s}{\mathrm{d}d_{\mathrm{p}}} = \frac{\mathrm{d}s}{\mathrm{d}d_{\mathrm{p}}} \int_{0}^{+\infty} K(s_{\mathrm{i}}, d_{\mathrm{p}}) \times \frac{\mathrm{d}N}{\mathrm{d}d_{\mathrm{p}}} \mathrm{d}d_{\mathrm{p}}$$

$$= \int_{d_{\mathrm{i}}-\delta}^{d_{\mathrm{i}}+\delta} K(s_{\mathrm{i}}, d_{\mathrm{p}}) \times \frac{\mathrm{d}N}{\mathrm{d}d_{\mathrm{p}}} \times \frac{\mathrm{d}s}{\mathrm{d}d_{\mathrm{p}}} \times \mathrm{d}d_{\mathrm{p}}$$

$$= \frac{\mathrm{d}N}{\mathrm{d}d_{\mathrm{p}}}(d_{\mathrm{i}}) \int_{s_{\mathrm{i}}-\varepsilon}^{s_{\mathrm{i}}+\varepsilon} K(s_{\mathrm{i}}, d_{\mathrm{i}}) \times \mathrm{d}s$$

$$= \frac{\mathrm{d}N}{\mathrm{d}d_{\mathrm{p}}}(d_{\mathrm{i}}) \int_{0}^{+\infty} K(s_{\mathrm{i}}, d_{\mathrm{i}}) \times \mathrm{d}s$$

$$= \frac{\mathrm{d}N}{\mathrm{d}d_{\mathrm{p}}}(d_{\mathrm{i}}) \times \eta(d_{\mathrm{i}}, s_{\max}),$$

where d is the derivate symbol; $R$ is the raw concentration; $s$ is the saturator flow rate; $K$ is the kernel; $d\mathrm{p}$ is the particle diameter; $s_{\mathrm{i}}$ and $d_{\mathrm{i}}$ are the saturator flow rate and diameter at the *i*th point, respectively; $\delta$ is Cauchy's definition of the limit; $\varepsilon$ is the error; $s_{\max}$ is the maximum saturator flow rate; $\eta$ is the overall detection efficiency; and $\int$ is the integral symbol. In the stepwise method, the resolution of the kernel is assumed to be positive infinity and hence, there is a one-to-one relationship between each saturator flow rate and the retrieved particle diameter.

[Figure]

Fig. S1. Comparison of the 6- (dotted) and 11-channel (solid) size bins. The grey line indicates the raw estimation, $R_{1.2\text{-}2.8}$ (calculated as difference between saturator flow rate at 1.3 lpm and 0.1 lpm). Top figures are during an NPF event (30 Jan.) and bottom figures are from a non-event day (6 Mar.).

[Figure]

Fig. S2. Inversion kernel using 6-channels. Note that these kernels were theoretically calculated and not implicitly used for the current study.

---

## Author Response (AR2)

**Author response to Referees**

**Associate Editor**

We have now added to the SI a step-by-step user guide on how to use the MATLAB code. In addition, the references section is now properly conformed (added DOIs and journals abbreviated).

**Referee 1**

- L14: „step" or „task" instead of "component"
  Changed to task
- L18: not clear what "event" means here
  Clarified the sentence
- L22: "failed" instead of "fared poorly"?
  "Failed" is a strong word. The stepwise and kernel methods did invert but not as consistent as the other methods. We would like to retain the wording, "fared poorly".
- L224 to L241: This section is quite confusing. As far as I understand it essentially says that 6 channels should be used but for some reason 11 channels are used in the example shown in Figure 1. Why did the authors not use 6 channels also in the example if it is the preferred number of size bins?
  As shown in the manuscript, using 6- or 11-channels makes no discernible difference. We wished to show this in Figs. 1 and S1 to illustrate this. However, we note that this study uses data from only one site. Therefore, as a precaution for general PSM users, using 6-channels would be more certain that the kernel function curves do not overlap. This is clearly explained why on L224-241
- L266, and L280 to L289: It seems "reasonable" in this case only means that the activation diameter decreases with increasing saturator flow rate? Is this correct.
  That is correct – this is explained in section 3.1 in more detail
- L285 to 286: Does this statement mean that the PSM does not give very accurate results when measuring ambient particles? If available, please cite the literature where this effect has been evaluated.
  No, the PSM does give accurate results. We put "may not be exceptionally accurate" in relation to the retrieval of the kernel function curves, which are always derived from laboratory conditions. We have clarified the sentence
- L336: Please explain what is meant by "background particle concentration". In addition, how is the "ratio" defined?
  The background concentration is concentration that is measured beyond the PSMs limits (i.e., n > 2.8 nm). We have added a sentence to define this term. The ratio has already been defined earlier in the manuscript (L253-255)
- L352 to L353: please explain how the "signal-to-noise ratio" of Figure 3 is defined
  As mentioned previously, the ratio is defined on L253-255. It may not be needed to restate it again
- L489 to L490: The link provided leads to an empty page.
  The link works fine. Likely because the link is over two lines, which may break the link

- Figure S2: It says in the caption "Note that these kernels were theoretically calculated and not implicitly used for the current study". Why? At least, the kernels that were used should be shown in comparison (e.g., in the same colors but as dashed lines).

    We have changed the Fig. S2 to show the actual kernel curve and not the theoretical (as to not confuse the reader). In addition, added the detection efficiency curve

[revised manuscript text omitted]

---

## Author Response (AR3)

**Author response to Referees**

**Associate Editor**

We have changed the abstract's first sentence as suggested by the Associate Editor. Additionally, in the manuscript, we have hyphenated two instances of "nanoparticles" to "nano-particles" to keep it consistent with this first sentence. No other changes were made.